# Survival-Larval Density Relationships in the Field and Their Implications for Control of Container-Dwelling *Aedes* Mosquitoes

**DOI:** 10.3390/insects14010017

**Published:** 2022-12-23

**Authors:** Katherine G. Evans, Zoey R. Neale, Brendan Holly, Cecilia C. Canizela, Steven A. Juliano

**Affiliations:** School of Biological Sciences, Illinois State University, Normal, IL 61790-4120, USA; kgevans@ilstu.edu (K.G.E.); zoey.neale@rice.edu (Z.R.N.); bholly24@gmail.com (B.H.); ceciliacanizela@gmail.com (C.C.C.)

**Keywords:** density dependent survival, mortality, overcompensation, larvicide, sterile insect technique, incompatible insect technique, population dynamics, *Aedes aegypti*, *Aedes albopictus*, *Aedes triseriatus*

## Abstract

**Simple Summary:**

*Aedes* mosquitoes with larvae that develop in water-filled containers are prominent vectors of disease and are often targets of mosquito control. Laboratory studies and theory suggest that survival of larvae to adulthood decreases as the density of larvae increases. Theory further suggests that this density effect may interact with mosquito control, such that mortality from mosquito control may sometimes increase production of adults. To understand whether such an effect is likely in nature, we conducted field studies of three *Aedes* species at five sites. Our field studies determined the typical range of densities in the field and quantified the shape of the survival-density relationship. We find that observed larval densities and survival-density relationships vary substantially, often resulting in the prediction that adult production would be unaffected, or even increased, by imposing mortality on larvae. Maximum larval density at a site is strongly and positively related to the likelihood of these counter-productive predicted outcomes. Our results indicate that we cannot assume that killing larvae will result in fewer adults. Effectiveness of mosquito control may be improved by a thorough understanding of how a local mosquito population will respond to achievable levels of larval mortality from mosquito control.

**Abstract:**

Population density can affect survival, growth, development time, and adult size and fecundity, which are collectively known as density-dependent effects. Container *Aedes* larvae often attain high densities in nature, and those densities may be reduced when larval control is applied. We tested the hypothesis that density-dependent effects on survival are common and strong in nature and could result in maximal adult production at intermediate densities for *Aedes aegypti*, *Aedes albopictus*, and *Aedes triseriatus*. We surveyed naturally occurring densities in field containers, then introduced larvae at a similar range of densities, and censused the containers for survivors. We analyzed the survival-density relationships by nonlinear regressions, which showed that survival-density relationships vary among seasons, sites, and species. For each *Aedes* species, some sites and times yielded predictions that larval density reduction would yield the same (compensation), or more (overcompensation), adults than no larval density reduction. Thus, larval control targeting these *Aedes* species cannot always be assumed to yield a reduction in the number of adult mosquitoes. We suggest that mosquito control targeting larvae may be made more effective by: Imposing maximum mortality; targeting populations when larval abundances are low; and knowing the shape of the survival-density response of the target population.

## 1. Introduction

Mosquitoes are common nuisance pests that can also vector disease. Mosquito-borne illnesses transmitted by *Aedes* species that inhabit small water-holding containers include yellow fever, Zika, dengue, and chikungunya [1]. Due to the threat *Aedes* mosquitoes pose as disease vectors, numerous mosquito control techniques have been developed to reduce populations of biting adult mosquitoes [1,2,3].

When controlling mosquitoes, ecological conditions must be considered to effectively reduce adult mosquito densities. Container-dwelling *Aedes* larvae are often subject to negative density-dependent effects, such that high densities of larvae may result in a high percentage of density-dependent mortality, reduced growth, and increased development time [4,5,6,7]. Conversely, low densities of *Aedes* larvae are likely to result in a low percentage of density-dependent mortality and relatively larger, rapidly developing individuals. Mosquito control typically involves reducing a population via extrinsic mortality, which is mortality imposed by a source outside of the population, such as harvest, predation, or human intervention. Laboratory studies indicate that extrinsic mortality of *Aedes* larvae that leaves some survivors in a high-density environment can result in a greater number of larvae surviving to adulthood than would survive if no extrinsic mortality had occurred, a phenomenon known as overcompensation [8,9,10]. Compensation occurs when the number of larvae surviving to adulthood is the same with or without extrinsic mortality [8].

Both compensation and overcompensation would be counter-productive results for mosquito management—where the goal is to reduce the number of biting mosquitoes. One hypothesized condition for overcompensation that has received theoretical and empirical support is that extrinsic mortality must act before the bottleneck induced by density-dependent mortality [9,11]. Additional theoretical work [12] suggests that mortality imposed on a population could result in increases of a specific life stage. Which life stage increases due to mortality depends on how the population is structured. If a bottleneck occurs at maturation in an uncontrolled population because of strong resource competition among juveniles, then extrinsic mortality will cause an increase of adults. 

Ecological theory on the effects of mortality on life stages is relevant to mosquito control, which typically imposes mortality on specific life stages (e.g., adulticides, larvicides, Sterile Insect Technique (SIT), Incompatible Insect Technique (IIT)) [13]. Abram’s [11] hypothesis predicts that larval control methods imposing mortality on early stages may be counter-productive because they are more likely to induce overcompensation. This hypothesis was supported by laboratory studies showing that early acting mortality of mosquito larvae resulted in overcompensation of adult production [9,10], but late-acting mortality of larvae did not, although even late acting mortality could result in compensation [9].

In recent years, public and commercial mosquito control programs have been moving toward species specific, pesticide-free approaches [2,3,14,15,16]. Two such approaches are the SIT and IIT, which have been used to target *Ae. albopictus* and *Ae. aegypti*. Both techniques release non-biting male mosquitoes incapable of siring offspring when mating with wild females. Females who mate with the modified males will lay eggs that never hatch. For our research, we consider SIT and IIT approaches to cause “early-stage” mortality because they reduce the number of larvae at the earliest possible point, i.e., before the larvae even hatch. In several published studies, the SIT and IIT approaches have been shown to significantly reduce the number of biting adults in a population compared to control sites, either separately [17,18], or combined [19]. However, the range of ecological conditions in which SIT and IIT will be effective has yet to be determined.

Two questions help us determine if overcompensatory or compensatory responses to mosquito control targeting larvae are likely under field conditions. (1) What is the typical range of densities of larvae that occur in the field? Additionally, (2) What is the relationship of survival to larval density in the field? (Figure 1). The shape of that relationship determines whether overcompensation or compensation are possible, as in the hump-shaped overcompensatory curve in Figure 1. Sauers et al. [20] showed that such hump-shaped curves, with maximal adult production at intermediate densities, can occur in container *Aedes* species in controlled laboratory conditions, but this relationship has rarely been quantified in the field. Beyond the shape of that relationship, the range of larval densities (i.e., the horizontal axis in Figure 1) also determines the response to any real implementation of mosquito control. All three relationships can result in additive mortality (a desirable result) if the density of larvae at the time mortality is imposed is below the peak of the hump for the overcompensatory relationship, or the asymptotic region of the compensatory relationship (Figure 1). Understanding the level and frequency of density-dependent mortality among mosquitoes can help inform control practices, particularly as species-specific control approaches targeting specific stages gain popularity. 

Previous field studies have indicated that density-dependence is likely to be strong in the field for container *Aedes* larvae at naturally occurring densities [4,5,6,21,22] but those studies were not designed to quantify the shape of the survival-density relationship, and provided only limited information on the typical densities occurring in the field. A reduction of some, but not all, larvae in containers, for example via SIT and IIT causing hatch reduction, may alleviate intraspecific resource competition among the survivors, but how populations might respond is not clear. 

In this paper we address both of those important questions—determining the shape of the survival-density relationship and the typical range of larval densities for three container *Aedes* species targeted by control efforts: *Ae. aegypti*, *Ae. albopictus*, and *Ae. triseriatus*. To address our hypothesis that overcompensation is likely to occur at field-relevant densities, the experiments spanned multiple sites in Florida and Midwest US across the summers of 2017–2019. Although overcompensation or compensation have been demonstrated empirically in controlled laboratory settings for these three species [9,10], field studies assessing the likelihood of these effects in container mosquitoes have yet to be conducted. Our research is a step toward understanding when and where early acting species-specific mosquito control techniques, such as SIT and IIT, can be implemented to effectively reduce mosquito populations.

## 2. Materials and Methods

### 2.1. General Approach

We conducted experiments on three container *Aedes* species: *Ae. triseriatus, Ae. albopictus*, and *Ae. aegypti* at five different sites (detailed site descriptions are in Appendix A). For each species, at least two experiments were run at different times, different field sites, or both different times and field sites. Our goal was to survey the range of larval densities at each site and determine the relationship between survival and larval density.

First, we conducted surveys (Table 1) of larval densities by setting out buckets or cemetery vases, depending on the sites, and filling them half-full of water. The containers were colonized by local *Aedes* for 4 to 6 weeks. After the survey, we removed the colonizers and any other aquatic invertebrates and counted total *Aedes* abundance. We excluded one container from analysis because it was colonized by *Culex* larvae at an extremely high density. 

After a survey period was complete, we carried out a density manipulation to determine the relationship between survival and larval density (Table 2). Because *Aedes* eggs are laid on container walls, we transferred the water and detritus from each survey container to a new, egg-free container for the next stage of the experiment. We added a cohort of first instar, laboratory-hatched *Aedes* larvae of one target species per site to each new container, which was a simplifying measure to avoid parsing effects of inter- vs. intra-specific competition when evaluating density-dependent effects. Colony rearing conditions and origins of the colonies are included in Appendix A. The numbers of first instar larvae that we added to new containers spanned the range of densities present at the site at the end of the survey period. Then, we sealed the containers with nylon mesh to eliminate further colonization and to trap any adults that eclosed. 

We began censusing containers 6 days after introducing the first-instar larvae to determine number of survivors and immature stages. We took a final census when a substantial number of pupae appeared in containers at a site. For the final census, we counted and collected the number of surviving larvae, pupae, and any adults. We analyzed the total number of survivors with a nonlinear regression (described below) to determine the shape of the relationship of survivors to larval density, which was used to infer the likelihood that overcompensation would occur at field-relevant densities. 

Variation in procedures among experiments includes the target species, date the survey started, site, length of the survey period, type of container, number of survey or experimental containers, water source for filling the container, initial numbers of experimental larvae, experimental census days, and whether or not litter was added during the experiment (Table 1 and Table 2).

### 2.2. Survey Period

The goal of the survey period for each experiment was to determine the densities of mosquito larvae that would naturally occur at a site with the introduction of new breeding containers. At the end of the survey period, we rinsed the water and material from each container through a 106 μm sieve (Thermo Fisher Scientific, Hampton, NH, USA).

The larvae from each container were removed, placed in labeled containers, and transported to the laboratory. We transferred the water and detritus from each container to a new container for the density manipulations. We then placed the new container back in the same location as the original container and sealed each new container to prevent oviposition by local populations. Buckets were sealed with plastic lids with a wire-mesh center covered with bridal veil (1.5 mm openings, sufficient to exclude or to confine adult mosquitoes); Cemetery vases were sealed by covering them with bridal veil and securing the veil with a rubber band. The lids allowed rainfall to accumulate in the containers, as well as airflow. Rainfall overflowed out of the top of the container if enough water accumulated. 

### 2.3. Density Manipulation

The larval numbers used in each experiment spanned the range of densities for *Aedes*-inhabited containers present during the survey period, with replication limited by the number of hatched larvae available. For example, if the survey yielded numbers of *Aedes* larvae ranging from 10–150 individuals, then numbers of first instar larvae added to the containers used would be 0, 10, 40, 80, 120, and 150 individuals (Table 2). 

We hatched eggs from lab-reared colonies of the target species, and after 24 h first-instar larvae were aliquoted into vials. We randomly assigned initial larval numbers to the experimental containers in the field. Then, we transported the vials of larvae to the field and added them to their assigned containers. Containers with 0 added larvae were used to determine if additional larvae beyond those we added were present in the containers. Numbers of larvae later found in containers with 0 added larvae were low, ranging from 0 to 23. Such contamination in the experimental containers could arise from incomplete removal of natural colonists, particularly first instar larvae, oviposition through the mesh covers during the experiment, dislodged eggs in the container water, or transfer of eggs along with detritus from secondary containers (described below). 

For some of the experiments, a secondary open-topped container of the same type was staked next to each experimental container. The secondary containers collected detritus during the experiment, which approximated what the experimental container would have accumulated if it had been uncovered. This collected detritus was added to the associated experimental containers on each census day. The secondary containers had concave wire mesh inserts, which kept detritus dry if rainfall accumulated. These secondary containers enabled us to test whether the sealing of our experimental containers, and resulting reduction in detritus accumulation, significantly reduced the resources available to larvae during the experiment. 

The experimental containers were censused for survivors on set days (Table 2). On each census day, detritus that had collected in the secondary containers was added to paired experimental containers. Pupae were collected into labeled vials and returned to the lab, where they were identified to species after eclosion. Adults that were present in the container were counted and identified to species (when identification was possible). The final census day was determined by the appearance of substantial pupae in the containers. On the final census day, we collected all remaining individuals—including larvae—and transported them to the laboratory.

The cumulative number of survivors was calculated as the total number of living mosquito larvae in a container on the final census day, plus any individuals that had been removed as pupae or counted as adults on previous census days. Individuals were identified to species, when possible, to ensure that contamination from the natural community was limited. In each experiment, individuals of the target species were assumed be the experimental individuals, we could not distinguish introduced individuals of the target species from contamination. Across all experiments, 4 experimental containers yielded more survivors of the target species than the number of experimental larvae, and these 4 replicates were omitted from analysis. Individuals of non-target species were not counted among survivors. 

### 2.4. Data Analysis

We used a version of the discrete time Shepherd [23] recruitment model (Equation (1)) [17,24,25] to quantify the relationship between survival and larval density. We used the parameter estimates from this model and the shape of the graph to predict whether overcompensation would be likely to occur at some of the natural densities, should extrinsic mortality have been imposed on cohorts of those densities.
*S* = *aN*/[1 + (*N/K*)*^d^*],(1)

This equation was used to determine the relationship between the initial density of larvae *N* and *S* the number of those individuals surviving. The parameters *a*, *K*, and *d* determine the shape of the relationship. Parameter *d* determines whether the shape of the relationship will increase monotonically, reaches an asymptote, or produce a hump-shaped curve (Figure 1). Parameter *a* estimates the density-independent survival of the larvae, i.e., the proportion of individuals predicted to survive as density approaches 0. Parameter *K* is the initial density at which the proportion of larvae surviving *S*/*N* = *a*/2. When the value of parameter *d* > 1.0, the survivors-density relationship is hump-shaped (Figure 1), and overcompensation is predicted to occur at high densities. When the value of parameter *d* = 1.0, the survivors-density relationship is asymptotic (Figure 1) and compensation is predicted to occur. When the value of parameter *d* < 1.0, the survivor-density relationship is monotonic increasing without an asymptote, and additive or sub-additive mortality is likely to occur at all densities. When *d =* 0, *S*/*N* is independent of *N* (i.e., all mortality is density independent) and number of survivors increases linearly with initial density (Figure 1).

The density manipulations varied in multiple ways (Table 2), hence each experiment was executed and analyzed separately by nonlinear regression of the cumulative number of survivors (*S*) on the final census day vs. initial density (*N*). A generalized nonlinear model approach (PROC NLMIXED, SAS 9.4, SAS Institute, Cary, NC, USA) with a Poisson distribution of error and a log link function was used to fit Equation 1 to each data set. We used 95% confidence intervals for the parameter *d* to test whether *d* was significantly greater or less than 1.0. ILForest18, MOForest18, FLSuburb18-1, and FLCemetery19-2 experiments had secondary detritus collection containers for half the container locations. A two-way fixed effects ANOVA was used to test whether detritus addition had any detectable effect or interaction with initial density (a class variable for this analysis) on the number of survivors. 

Across all experiments and sites, we tested the hypothesis that we are more likely to predict overcompensation (i.e., value of *d* > 1) for experiments characterized by greater maximum density in the survey period. That hypothesis predicts that greater peak density of *Aedes* larvae at a site is positively associated with greater estimated *d*. We performed a simple one-tailed linear regression to test the prediction that estimated *d* increases with the maximum number of larvae per container during the survey period. Linear regression and assumption testing were carried out using Excel with Real Statistics Resource Pack software (Release 7.6, 2013–2021) (https://www.real-statistics.com/).

## 3. Results

### 3.1. Survey Results and Density Manipulation 

ILForest17—The frequency distribution of the number of larvae per container at the end of the survey period (Figure 2A) had a minimum of 14 larvae in a container and maximum of 498. The primary species that colonized the containers were *Ae. japonicus* and *Ae. triseriatus*. The estimated parameter value of *d* was significantly greater than 1 (Table 3). Therefore, overcompensation would be predicted at this site at the time of this experiment for *Ae. triseriatus*. Specifically, overcompensation would have been likely at initial densities from the higher end of the survey range (Figure 2B). Our data and corresponding model predictions indicate fewer survivors at an initial larval density of 320 compared to initial densities from 40 to 160 (Figure 2B).

ILForest18—The frequency distribution of the number of larvae per container at the end of the survey period (Figure 2C) had a minimum of 3 larvae in a container and maximum of 298. Similar to the previous year, the primary species that colonized the containers were *Ae. japonicus* and *Ae. triseriatus*. The estimated parameter value of *d* was significantly less than 1, (Table 3). Therefore, overcompensation would not be predicted at this site over the course of this experiment for *Ae. triseriatus*, and additive mortality would be expected at the observed densities. Our data and corresponding model predictions show a monotonic increase in the number of survivors as the initial larval density increases (Figure 2D). 

MOForest18—The frequency distribution of the number of larvae per container at the end of the survey period (Figure 3A) had a minimum of 2 larvae in a container and maximum of 744. The primary species that colonized our survey containers during the survey period were *Ae. japonicus* and *Ae. triseriatus*, similar to ILForest17 and ILForest18 experiments. The density manipulation was carried out on *Ae. albopictus*. The parameter value of *d* was significantly greater than 1 (Table 3). Therefore, overcompensation would be predicted at this site during this experiment on *Ae. albopictus*. Our data and corresponding model predictions indicate that overcompensation would have been likely to occur at densities >400 *Aedes* larvae, which we interpret from the low number of survivors with an initial larval density of 500 compared to initial densities of ≤400 larvae (Figure 3B). 

ILCemetery19—The frequency distribution of the number of larvae per container at the end of the survey period (Figure 3C) had three containers with 0 larvae and a maximum number of larvae of 198. The primary species to colonize the containers during the survey period were *Ae. japonicus*, *Ae. triseriatus*, and *Ae. albopictus*. The parameter value of *d* was significantly less than 1 (Table 3). Therefore, overcompensation would not be predicted to occur at this site over the course of the experiment. Our data and corresponding model predict that additive mortality would be likely at the densities observed for this experiment (Figure 3D).

FLSuburb18-1—The frequency distribution of the number of larvae per container at the end of the survey period (Figure 4A) had a minimum of 1 *Aedes* larva and a maximum of 457. The parameter value of *d* was not significantly different from 1 (Table 3), thus predicting compensation may have been likely at this site during the experiment. Our data and model predictions indicate that the number of survivors increases as the initial larval density increases for the majority of the densities tested, but the slope for the predicted number of survivors begins to flatten out at the highest density treatment of 350 (Figure 4B). Even excluding one container colonized by *Culex*, the greatest naturally occurring densities during the survey period were higher than those tested during the experiment (Figure 4A,B). Based on the parameter *d* value and the range of densities observed during the survey, we conclude that compensation could have been likely to occur at this site for *Ae. aegypti* at densities higher than those tested in the density manipulation experiment.

FLSuburb18-2—The frequency distribution of the number of larvae per container at the end of the survey period (Figure 4C) had a minimum of 22 larvae in a container and maximum of 196. Despite the lower densities compared to FLSuburb18-1 (compare Figure 4A–D), the parameter value of *d* was again not significantly different from 1 (Table 3). Therefore, compensation would have been predicted at this site for *Ae. aegypti*, as indicated by the regression predictions showing an approximately equal number of survivors for initial larval densities of ≥100 (Figure 4D).

FLCemetery19-1—This was the only experiment in which we used established containers (in this case, cemetery vases not placed in the field by us) rather than newly placed experimental containers for the survey to determine naturally occurring densities. *Aedes aegypti* were the primary colonizers in these established containers. The frequency distribution of the number of larvae per container (Figure 5A) had five containers with 0 larvae and a maximum number of larvae of 89. The parameter value of *d* was significantly less than 1 (Table 3). Therefore, overcompensation would not be predicted to occur for *Ae. aegypti* in the established containers at this site during the experiment. Our data and model predictions indicate that additive mortality would be likely across all densities, as indicated by the strongly linear increase in the number of survivors as the initial larval density increases (Figure 5B).

FLCemetery19-2—This experiment took place at the same location as FLCemetery19-1, but included a survey period using new experimental containers. Similar to the established containers in FLCemetery19-1, the primary species that colonized the containers was *Ae. aegypti*. The frequency distribution of the number of larvae per container at the end of the survey period (Figure 5C) had eight containers with 0 larvae and a maximum number of larvae in a container of 161. The parameter value of *d* was not significantly different from 1 (Table 3). Our data and model predictions indicate that compensatory mortality would occur at the higher densities observed during the survey period. 

### 3.2. Detritus Effect

All four experiments yielded no significant effect of detritus addition on adult production, and only the ILForest18 experiment yielded a marginally significant interaction of detritus addition and initial density (Appendix A, Table A1). This significant result was likely driven by a large difference in survivors at highest initial density of 300 larvae, where the treatment with added litter yielded 104 survivors compared to the no litter treatment that yielded 34 survivors (Figure 2D).

### 3.3. Regression 

Regression of estimated *d* vs. the maximum number of larvae observed during the survey (Figure 6) met the assumptions of linearity and homogeneity of variance. Results of the linear regression analysis indicate a significant positive relationship between the maximum number of larvae at a site and the value of parameter *d* (one tailed t_1,6_ = 3.32, p = 0.0080), which is consistent with our hypothesis that we are more likely to predict overcompensation when larval density in the survey period is greater.

## 4. Discussion

Our experiments addressed two important questions: (1) What is the typical range of densities of larvae that occur in the field? Additionally, (2) What is the relationship of survival to larval density in the field? By asking these questions, we tested the hypothesis that density-dependent effects are strong and common in the field, and sufficient to result in overcompensatory responses to control efforts. Our experiments investigated larval densities for three North American container-dwelling *Ae. aegypti*, *Ae. albopictus*, and *Ae. triseriatus*, all important vectors of disease. Our results indicate that if extrinsic mortality had occurred, it would have been likely to result in either compensation or overcompensation in 5 of 8 experiments, and in at least one experiment for each species. Field studies, such as this one, provide empirical evidence that phenomena observed in laboratories or predicted by theory also do occur in nature. Our research therefore provides evidence for density-dependent effects on survival among *Aedes* larvae in nature, which is likely to be important information for improving practices in vector control.

Overall, our results indicate that a reduction in number of survivors cannot be uniformly assumed when larval control methods are used to reduce larval populations of these *Aedes* species in the field. Our experiments further showed that the likelihood of overcompensation and compensation is associated with environmental variation, as at least one of the experiments for each species indicated that only additive mortality would have occurred with the application of extrinsic mortality. Thus, we conclude that the effects of mosquito control of these container *Aedes* are likely to vary among sites, times, and species.

Results from our regression analysis across the experiments suggest that estimates of parameter *d* were quite strongly positively associated with maximum numbers of larvae, with about 68% of the variation in the value of parameter *d*, which controls the shape of the survivors-density relationship, explained by the maximum number of *Aedes* larvae per container during the survey. While the number of larvae present at the site is clearly a strong factor in predicting whether or not overcompensation may occur, a simple measurement such as this is likely not sufficient for predicting whether or not counter-productive results are likely. The relationship between density and survivor number could be further explained by environmental differences caused by locations, seasonal timing, year-to-year variation, and newly placed versus established containers at a site. Most of the evidence from our experiments supplementing containers with litter that would have accumulated during the experiment indicates that any litter effects during these short-term experiments were small, and only appeared to affect survivor number at one site, and at the highest experimental density. Nevertheless, we expect that variation in detritus resources may be another important determinant of responses to control efforts outside of our experimental context.

Factors that are known to influence intraspecific competition among mosquito larvae include detritus level and type (e.g., [26,27,28,29]), temperature (e.g., [30,31,32]), predator presence (e.g., [33,34]), and container drying (e.g., [35]). Our experiments show variation in density dependent effects on survival, but we did not explicitly test which environmental factors may be causing such variation in density-dependent effects. Future work should explore the factors, or combination of factors, that may affect the relationship of survivors to larval density for container *Aedes*. 

We have shown that the shape of the relationship between *Aedes* larval survival and larval density differs among environmental contexts (site, seasonal time, species, range of larval densities). The shapes of those relationships are likely to influence the impact of control efforts on adult production. Determining how environmental factors affect that shape in the field will therefore be important for predicting the success of mosquito control programs. For example, we predict greater resources to shift an asymptote or hump-shaped curve further right (Figure 7). Environments with greater resources will support more individuals surviving to adulthood, and thereby lessen the negative-density dependent effects across a wider range of densities. We further predict that imposing mortality in a high resource environment would be more likely to result in additive mortality across a wider range of densities than in a low resource environment, as shown by compensatory and overcompensatory relationships diverging from additive at a greater initial density in a high resource environment (Figure 7). We postulate that greater resource accumulation in the established vases in FLCemetery19-1 may have contributed to reduced competition among larvae, leading to a lower magnitude of density-dependent effects (Figure 5). Notably, many vases had accumulated live oak leaves, which indicates that the vases had been accumulating resources for at least five months, as live oak leaves fall abundantly from January to March [36]. Another contributing factor to the lesser impact of density on survival in FLCemetery19-1 is likely the generally low densities of larvae observed in the survey period (Figure 5A).

While our research focused on intraspecific density dependence via competition, interspecific competition is another important consideration for regions where two or more *Aedes* species co-occur. Control methods such as SIT and IIT are species-specific, and therefore may have the potential to change interspecific competition dynamics by targeting one species but not the other. However, even less targeted pesticides can alter interspecific competition among *Aedes*, likely by differentially impacting the competitors. One laboratory study found that compensation of *Ae. aegypti* adult production occurred in response to malathion in the presence of *Ae. albopictus* [37]. Further research on how larval control methods affect intraspecific competition—as investigated here—and interspecific competition between *Ae. albopictus* and *Ae. aegypti* will be necessary to avoid having species-specific control methods produce counter-productive outcomes for one or both populations in regions of cooccurrence. 

Mosquito control efforts, particularly those that primarily or exclusively kill early-stages, will be most effective if the conditions that could produce overcompensation are identified and avoided. Some general principles for enhanced effectiveness may include: (1) Imposing the maximum possible extrinsic mortality (i.e., ensuring that the “Resulting new density” shown in Figure 1 and Figure 7 is as low as possible); (2) Targeting populations when larval abundances are low, so that negatively density-dependent effects on survival are unlikely (i.e., imposing mortality on populations at the low end of the horizontal axis in Figure 1 and Figure 7); (3) Knowing the shape of the survival-density response of the target species at the time and place control will be implemented (i.e., which of the relationships in Figure 1 or Figure 7 is most likely at that time and place). Year to year environmental differences in precipitation frequency and level [38], seasonal fluctuations [39], and container and detritus age [40] could all contribute to the degree of resource competition experienced by *Ae. aegypti*, *Ae. albopictus*, and *Ae. triseriatus*. Laboratory studies indicate that the combination of low detritus level and high larval abundance are likely to yield high density-dependent mortality [10] and make overcompensatory responses more likely. While modeling and laboratory studies can help narrow the factors that lead to high abundance and low detritus conditions for container breeding mosquitoes, further empirical studies at field-relevant densities and sites are necessary for better understanding and predictions of the outcomes of mosquito control.

## Figures and Tables

**Figure 1 insects-14-00017-f001:**
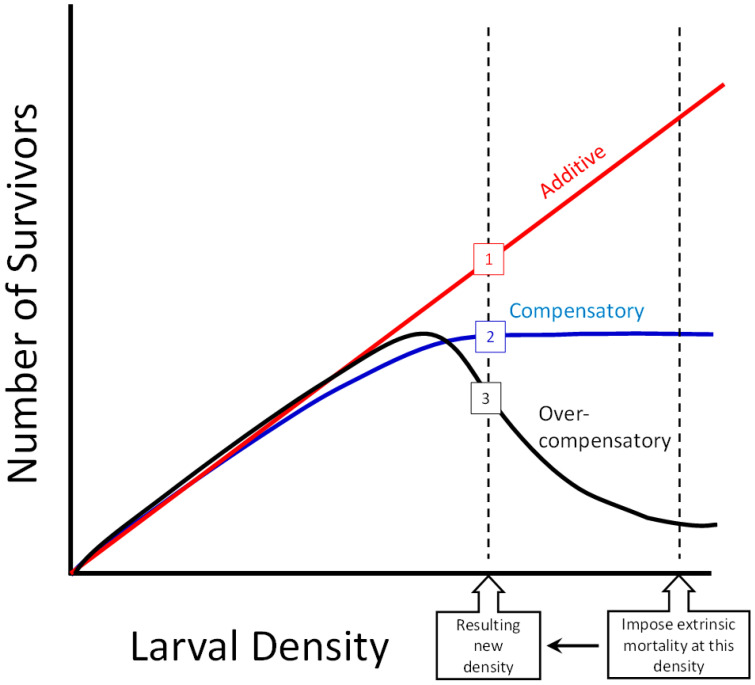
Three potential relationships between initial larval density and number of survivors, and the effect of imposing extrinsic mortality on a high-density population, reducing its density to a new, lower population density. Additive (red): imposing mortality would result in fewer survivors compared to not imposing mortality, as shown by point (1). Compensatory (blue): imposing mortality would result in the same number of survivors compared to not imposing mortality, as shown by point (2). Overcompensatory (black): imposing mortality would produce a greater number of survivors compared to not imposing mortality, as shown by point (3).

**Figure 2 insects-14-00017-f002:**
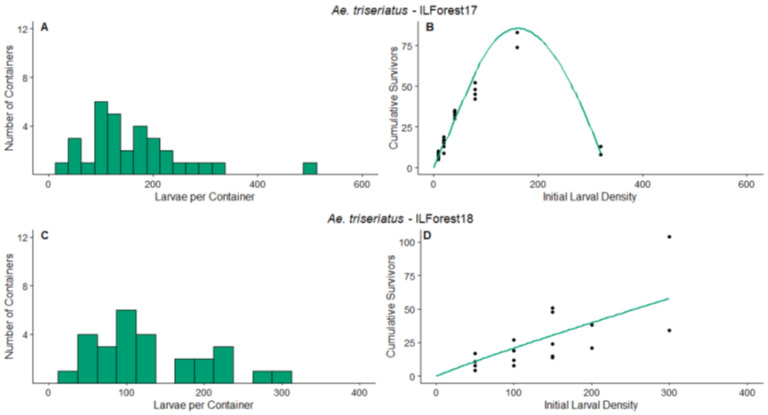
Results for ILForest17 and ILForest18 experiments. The left panels are the frequency distribution of mosquito larvae per container at the end of the survey periods, (**A**) for ILForest17, (**C**) for ILForest18. The points on the right panels represent the cumulative number of survivors on the final census day of the density manipulation experiments, (**B**) for ILForest17, (**D**) for ILForest18. The lines on the right panel represent the predicted values for the relationship between initial larval density and cumulative survivors, according to the non-linear regression analyses. Green indicates that the experiment was conducted on *Ae. triseriatus*. Axis values vary between experiments.

**Figure 3 insects-14-00017-f003:**
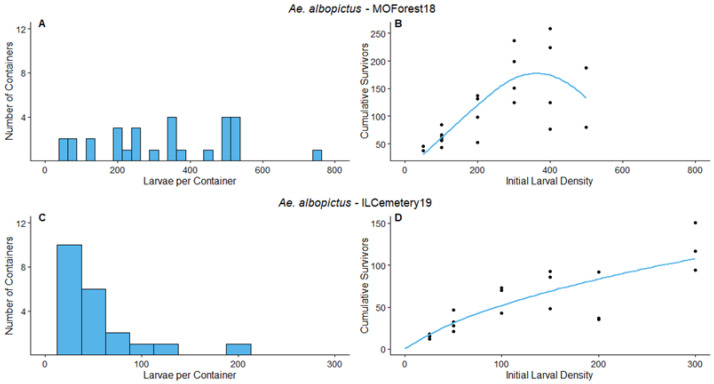
Results for MOForest18 and ILCemetery19 experiments. The left panels are the frequency distribution of mosquito larvae per container at the end of the survey periods, (**A**) for MOForest18, (**C**) for ILCemetery19. The points on the right panels represent the cumulative number of survivors of the density manipulation experiments, (**B**) for MOForest18, (**D**) for ILCemetery19. The lines on the right panel represent the predicted values for the relationship between initial larval density and cumulative survivors, according to the non-linear regression analyses. Blue indicates that the experiment was conducted on *Ae. albopictus*.

**Figure 4 insects-14-00017-f004:**
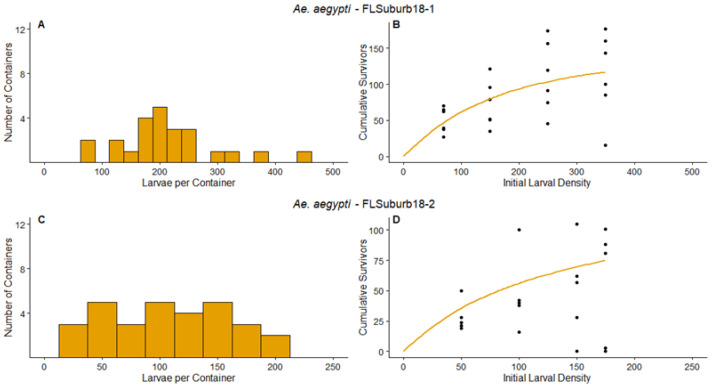
Results for FLSuburb18-1 and FLSuburb18-2 experiments. The left panels are the frequency distribution of mosquito larvae per container at the end of the survey period, (**A**) for FLSuburb18-1, (**C**) for FLSuburb18-2. (Not pictured—one container with 1055 *Culex* larvae in FLSuburb18-1). The points on the right panels represent the cumulative number of survivors of the density manipulation experiments, (**B**) for FLSuburb18-1, (**D**) for FLSuburb18-2. The lines on the right panel represent the predicted values for the relationship between initial larval density and cumulative survivors, according to the non-linear regression analyses. Orange indicates the experiment was conducted on *Ae. aegypti*. Axis values vary between experiments.

**Figure 5 insects-14-00017-f005:**
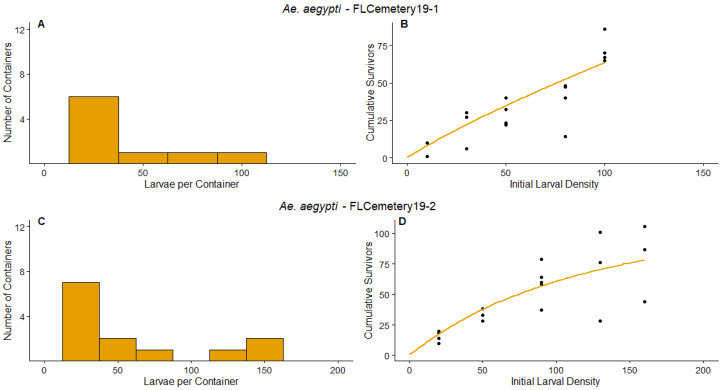
Results for FLCemetery19-1 and FLCemetery19-2 experiments. The left panels are the frequency distributions of mosquito larvae per container at the end of the survey periods, (**A**) for FLCemetery19-1, (**C**) for FLCemetery19-2. The points on the right panels represent the cumulative number of survivors of the density manipulation experiments, (**B**) for FLCemetery19-1, (**D**) for FLCemetery19-2. The lines on the right panels represent the predicted values for the relationship between initial larval density and cumulative survivors, according to the non-linear regression analyses. Orange indicates the experiment was conducted on *Ae. aegypti*. Axis values vary between experiments.

**Figure 6 insects-14-00017-f006:**
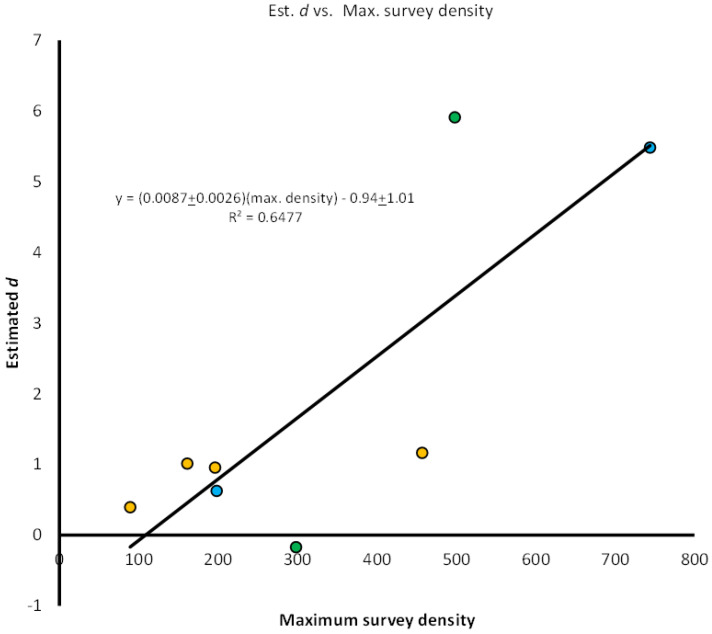
Regression of estimated value of parameter *d* plotted against maximum number of larvae in a container for each experiment. *d* = −0.94 ± 1.01 + 0.0087 ± 0.0026 (Max. survey density) *r*^2^ = 0.6477. Point colors indicate target species, and reproduce the colors used in Figure 2, Figure 3, Figure 4 and Figure 5: Green = *Ae. triseriatus*; Blue = *Ae. albopictus*; Orange = *Ae. aegypti*.

**Figure 7 insects-14-00017-f007:**
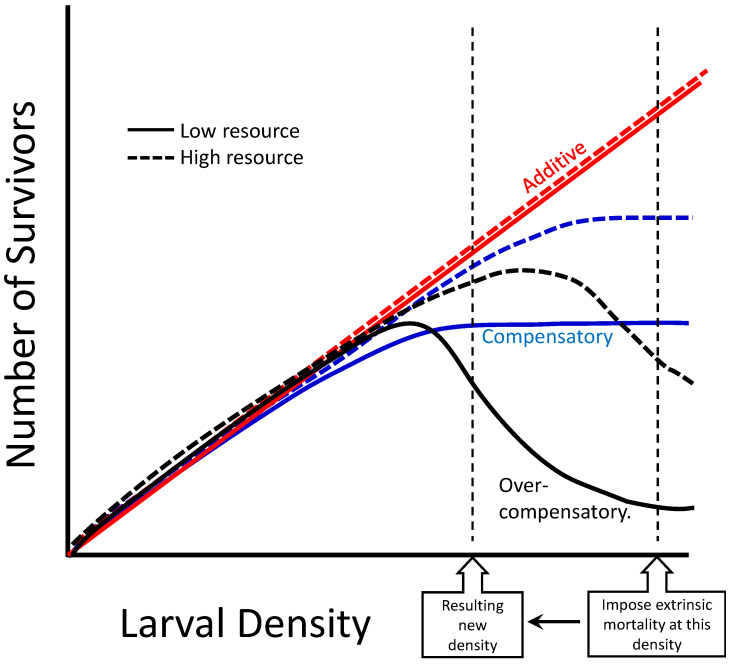
Postulated effects of resource levels on three relationships between initial larval density and number of survivors. Dotted and solid curves of the same color represent the postulated relationships in a relatively higher vs. lower resource environment, respectively. Additive (red): greater resource levels will not alter the result that imposing mortality would produce fewer survivors compared to not imposing mortality. Compensatory (blue): greater resource levels result in a greater asymptotic number of survivors, and could move the range of densities at which compensation occurs to greater values on the horizontal axis. Overcompensatory (black): greater resource levels result in a greater peak number of survivors and moves the density yielding that number to greater values on the horizontal axis. Imposing mortality in a higher resource environment may then produce a lesser magnitude of overcompensation relative to that produced in a low resource environment.

**Table 1 insects-14-00017-t001:** Methods for experiment sampling periods. The variations in procedures between experiments for the survey period are outlined above. The experiments are grouped by the target species. 1 L vases are cemetery vases fixed to the ground by a stake. Rain barrel water was collected from barrels on-site and sieved to remove larvae, predators, and parasites. N/A = not applicable, because we sampled established cemetery vases.

Experiment Name	Target Species	Start Date	Experimental Site	Location	Survey Period (Weeks)	Number of Survey Containers	Container Type	Water
ILForest17	*Ae. triseriatus*	June 2017	Merwin Nature Preserve	McLean County, IL, USA	6	32	8 L bucket	Rain barrel, sieved
ILForest18	*Ae. triseriatus*	June 2018	Merwin Nature Preserve	McLean County, IL, USA	6	30	4 L bucket	Rain barrel, sieved
MOForest18	*Ae. albopictus*	June 2018	Tyson Research Center	St. Louis County, MO, USA	6	30	4 L bucket	Rain barrel, sieved
ILCemetery19	*Ae. albopictus*	June 2019	Graceland/Fairlawn Cemetery	Decatur, IL, USA	6	27	1 L vase	Distilled water
FLSuburb18-1	*Ae. aegypti*	June 2018	Vero Beach Suburb	Vero Beach, FL, USA	4	26	4 L bucket	Distilled water
FLSuburb18-2	*Ae. aegypti*	July 2018	Vero Beach Suburb	Vero Beach, FL, USA	4	30	4 L bucket	Distilled water
FLCemetery19-1	*Ae. aegypti*	June 2019	Memorial Gardens Cemetery	Fort Myers, FL, USA	N/A	22	1 L vase	Rainwater
FLCemetery19-2	*Ae. aegypti*	June 2019	Memorial Gardens Cemetery	Fort Myers, FL, USA	4	29	1 L vase	Distilled water

**Table 2 insects-14-00017-t002:** Methods for density manipulation experiments. The variation in procedures among experiments for the density manipulations are outlined above. The experiments are grouped by the target species. Detritus additions are from secondary containers to supplement resources throughout the experiment. “Half” indicates that half of the containers received detritus additions. Census days are relative to the start date.

Experiment Name	Target Species	Start Date (day 0)	Container Type	Initial Larval Densities (Number of Containers)	Census Days	Detritus Addition
ILForest17	*Ae. triseriatus*	21 July 2017	8L bucket	10(6) 20(6) 40(4) 80(4) 160(2) 320(2)	6, 10, 14, 18	No
ILForest18	*Ae. triseriatus*	11 July 2018	4L bucket	50(4) 100(4) 150(6) 200(2) 300(2)	6, 10, 14, 18	Half
MOForest18	*Ae. albopictus*	23 June 2018	4L bucket	50(2) 100(8) 200(4) 300(4) 400(4) 500(2)	6, 10, 14	Half
ILCemetery19	*Ae. albopictus*	28 June 2019	1L vase	25(4) 50(4) 100(4) 150(3) 200(3) 300(3)	6, 9, 12	All
FLSuburb18-1	*Ae. aegypti*	30 June 2018	4L bucket	70(6) 150(6) 250(6) 350(6)	6, 9, 12	Half
FLSuburb18-2	*Ae. aegypti*	1 August 2018	4L bucket	50(5) 100(5) 150(5) 175(5)	6, 9, 13	All
FLCemetery19-1	*Ae. aegypti*	22 July 2019	1L vase	10(4) 30(4) 50(4) 80(4) 100(4)	6, 9, 12, 15, 17	All
FLCemetery19-2	*Ae. aegypti*	22 July 2019	1L vase	20(4) 50(3) 90(5) 130(3) 160(3)	6, 9, 12, 15, 17	Half

**Table 3 insects-14-00017-t003:** Density manipulation experiment results. The estimated parameter values for the non-linear regression (Equation (1)) are reported for each experiment, respectively. The interpretations of the results are based on the parameter *d* and are reported in the last column. *d* < 1 additive mortality; *d* = 1 compensation; *d* > 1 overcompensation. N/A indicates that confidence intervals for the associated parameter could not be estimated, usually because the solution converged to an estimate of the upper bound for the parameter (i.e., FLSuburb18-2, FLCemetery19-1, and FLCemetery19-1 experiments).

Experiment Name	Target Species	*a*	95% CI	*K*	95% CI	*d*	95% CI	Interpretation
ILForest17	*Ae. triseriatus*	0.663	[0.596, 0.731]	187.4	[169.0, 205.9]	5.9096	[4.6543, 7.1649]	Overcompensation
ILForest18	*Ae. triseriatus*	0.445	[−0.816, 1.707]	500	[−15,885, 16,885]	−0.1693	[−0.6292, 0.2906]	Additive mortality
MOForest18	*Ae. albopictus*	0.605	[0.5669, 0.6426]	476.8	[457.0, 496.7]	5.4821	[3.8778, 7.0865]	Overcompensation
ILCemetery19	*Ae. albopictus*	0.944	[0.104, 1.283]	137.2	[−272.0, 546.5]	0.6269	[−0.0295, 1.2833]	Additive mortality
FLSuburb18-1	*Ae. aegypti*	0.826	[0.466, 1.186]	250.9	[48.4, 451.6]	1.1658	[0.4167, 1.9150]	Compensation
FLSuburb18-2	*Ae. aegypti*	0.635	[0.379, 0.890]	167.4	[95.7, 239.8]	1.8690	[0.0219, 3.7161]	Compensation
FLCemetery19-1	*Ae. aegypti*	1	N/A	400.2	[−222.5, 1023.0]	0.3976	[0.03373, 0.7614]	Additive mortality
FLCemetery19-2	*Ae. aegypti*	1	N/A	153.8	[122.8, 184.9]	1.0134	[0.5263, 1.5005]	Compensation

## Data Availability

After publication, supporting data will be available on the Figshare site for Steven Juliano.

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
