# Peer review of "Survival-Larval Density Relationships in the Field and Their Implications for Control of Container-Dwelling Aedes Mosquitoes"

_insects, 2022, doi:10.3390/insects14010017_

Round 1

Reviewer 1 Report

This review is interesting, well written and organized. The authors addressed two questions: 1) What is the typical range of densities of larvae that occur in the field? And 2) What is the relationship of survival to larval density in the field? It’s helpful to the scientists working on the mosquito-borne diseases. However, there are still some linguistic mistakes in the manuscript, the authors should proofread thoroughly before it can be published.

Minor suggestions:

1. The number of words in the abstract exceeds 200. According to the requirements of Insects, please limit the abstract to 200 words.

2. Please use Abbreviated Journal Name in the References

Author Response

This review is interesting, well written and organized. The authors
addressed two questions: 1) What is the typical range of densities
of larvae that occur in the field? And 2) What is the relationship of
survival to larval density in the field? It’s helpful to the scientists
working on the mosquito-borne diseases. However, there are still
some linguistic mistakes in the manuscript, the authors should
proofread thoroughly before it can be published.

RESPONSE:  We have proofed the ms and made some minor corrections in addition to responses to specific comments. 

Minor suggestions:
1. The number of words in the abstract exceeds 200. According to
the requirements of Insects, please limit the abstract to 200 words.

RESPONSE: rewording of the abstract yields a word count of 199.

2. Please use Abbreviated Journal Name in the References

RESPONSE:  Done.  Let us know if we have used incorrect abbreviations.

Reviewer 2 Report

This is a clearly-written manuscript that demonstrated survival-density relationship of 3 Aedes species using field-based studies. Just some comments that would be useful to clarify the manuscript further:

L141, L163: Appendix 1 should be Appendix A.

L155, L172: Not sure what "1 Tables may have a footer" refers to.

Table 2: It would be good to provide the breakdown of the sample size/replicates for the respective larval densities. For instance in ILForest17, how were the densities (10, 20, 40, 80, 160, 320) distributed among the 28 experimental containers? And how would these distributions relate to the histograms in Figs 2-5?

L234: The authors should provide more information on the contamination and how it would or would not influence S (number of individuals surviving) and the results.

L234-236: Grammatical error

L371, L390, L526: Italicised species "Ae. aegypti"

L404: Italicised "d"

Figs. 1 & 7: They are largely similar and should be combined.

L608: The abbreviated genus name should be "Ae." instead of "A."

References: Should standardise to either sentence or title case.

Author Response

Note:  Line Numbers refer to the tracked version

L141, L163: Appendix 1 should be Appendix A.

RESPONSE:  Done

L155, L172: Not sure what " Tables may have a footer" refers to.

RESPONSE:  Removed.  This was part of the template that we neglected to remove

Table 2: It would be good to provide the breakdown of the sample
size/replicates for the respective larval densities. For instance in
ILForest17, how were the densities (10, 20, 40, 80, 160, 320)
distributed among the 28 experimental containers?

RESPONSE:  These are now provided in Table 2 for each of the experimental densities in each experiment.   As this renders the  column for total number of experimental containers redundant, that column has been removed. 

And how would these distributions relate to the histograms in Figs 2-5?

RESPONSE:  As we noted (current line 258) experimental densities were assigned at random, hence were approximately uncorrelated with colonist densities shown in the figures.

L234: The authors should provide more information on the
contamination and how it would or would not influence S (number
of individuals surviving) and the results.

RESPONSE:  We have added comments on this in the Methods (Lines 262-263 and also lines 295-298).  This is a good point and it has led us to delete a total of 4 observations in which a container yielded more survivors of the target species than the manipulated density.  This led to some small changes in the nonlinear regressions, which are now reflected in the parameters in Table 3, Figs. 2-5, and in the regression in fig. 6.  None of these small changes has altered our main conclusions.  We note that any undetected contamination with the target species may well reduce the likelihood of us detecting overcompensation, particularly if they occur at the highest densities. 

L234-236: Grammatical error

RESPONSE:  I believe we corrected this (current lines 266-271) but it not exactly clear from the comment what the grammatical error is.

L371, L390, L526: Italicised species "Ae. aegypti"

RESPONSE:  Done throughout

L404: Italicised "d"

RESPONSE:  Done

Figs. 1 & 7: They are largely similar and should be combined.

RESPONSE:  The two figures have different purposes - Fig. 1 to illustrate how shapes relate to overcompensation / compensation / additivity in response to density reduction, illustrated by the points labelled 1, 2, 3.  
In contrast fig. 7  illustrates hypothetical responses of the curves to greater / lesser resource availability (the solid / dashed curves).  We think showing both of those things in one graph would make the graph complex and confusing. 

We prefer the figures to be separate as we don't discuss Fig. 7 until the discussion, but we need to show fig. 1 in the introduction to explain how the shape of the survivors curve relates to overcompensation in response to density reduction.

We could combine the two figures into a 2 panel figure; however this is somewhat awkward, as we do not discuss figure 7 until the discussion section of the paper.

L608: The abbreviated genus name should be "Ae." instead of "A."

RESPONSE:  Done

References: Should standardise to either sentence or title case.

RESPONSE: Standardized to sentence case